# A new analysis method for evaluating bacterial growth with microplate readers

**Venkata Rao Krishnamurthi[1], Isabelle I. Niyonshuti[2], Jingyi Chen[2,3], Yong Wang[1,3,4]***

**1** Department of Physics, University of Arkansas, Fayetteville, AR, United States of America, **2** Department of Chemistry and Biochemistry, University of Arkansas, Fayetteville, AR, United States of America, **3** Materials Science and Engineering Program, University of Arkansas, Fayetteville, AR, United States of America, **4** Cell and Molecular Biology Program, University of Arkansas, Fayetteville, AR, United States of America

* yongwang@uark.edu

**Data Availability Statement:** All relevant data are within the manuscript and its Supporting Information files.

**Funding:** YW Grant No. ABI-0189, No. ABI-0226, No. ABI-0277, No. ABI-0326, No. ABI-2021

## Abstract

Growth curve measurements are commonly used in microbiology, while the use of microplate readers for such measurements provides better temporal resolution and higher throughput. However, evaluating bacterial growth with microplate readers has been hurdled by barriers such as multiple scattering. Here, we report our development of a method based on the time derivatives of the optical density (OD) and/or fluorescence (FL) of bacterial cultures to overcome these barriers. First, we illustrated our method using quantitative models and numerical simulations, which predicted the number of bacteria and the number of fluorescent proteins in time as well as their time derivatives. Then, we systematically investigated how the time derivatives depend on the parameters in the models/simulations, providing a framework for understanding the FL growth curves. In addition, as a demonstration, we applied our method to study the lag time elongation of bacteria subjected to treatment with silver (Ag$^+$) ions and found that the results from our method corroborated well with that from growth curve fitting by the Gompertz model that has been commonly used in the literature. Furthermore, this method was applied to the growth of bacteria in the presence of silver nanoparticles (AgNPs) at various concentrations, where the OD curve measurements failed. We showed that our method allowed us to successfully extract the growth behavior of the bacteria from the FL measurements and understand how the growth was affected by the AgNPs.

## Introduction

Growth curve measurement based on optical density (OD) is one of the most commonly used methods in microbiology for monitoring the growth and proliferation of microbes in time, which provides a simple, reliable and routine way to understand various aspects of the microbes [1–4]. For example, it has been used to routinely determine the growth of bacteria and other microbes treated with antibiotic drugs and substances [5–9], to study the responses of microbes to various environmental changes and stresses [10–12], and to monitor the accumulation of biomass during fermentation [13–15]. Traditionally, the growth curve measurements are performed by measuring the OD of the bacteria, which is related to the cell number, in cuvettes at the wavelength of 600 nm using photometry at desired time points with intervals

Arkansas Biosciences Institute https://arbiosciences.org/ YW & JC Grant No. 1826642 National Science Foundation https://nsf.gov/awardsearch/showAward?AWD_ID=1826642&HistoricalAwards=false.

**Competing interests:** The authors have declared that no conflict of interest exist.

of 30–60 min [3, 4]. Recent versions take advantages of automation and parallel measurements to achieve better temporal resolution and higher throughput. For example, turbidostats (or chemostats or morbidostats) have been developed and used for this purpose [16–18]. In addition, microplate readers with 96- or 384-well plates have been increasingly useful for monitoring the growth of microbes [19] and investigating the antimicrobial activity and mechanism of various drugs and substances. For example, microplate readers have been used to screen several thousand of *E. coli* mutants with single-gene deletions and provide detailed insight into how the bacteria respond to different silver nanoparticles (AgNPs) [20, 21]. In addition to better temporal resolution and higher throughput, another advantage of the microplate readers is that they usually support multimode measurements, providing a convenient way to monitor fluorescence (FL) simultaneously with the OD [22, 23].

Despite the increasing usefulness, quantitative applications of microplate readers for monitoring microbial growth have been hurdled by several barriers and issues. First, multiple scattering of microbes is usually severe and problematic in microplate reader-based OD measurements, as microplate readers have been predominantly used at higher culture densities [2]. As an example, a representative OD curve of an *E. coli* culture from a 96-well microplate reader is shown in Fig 1A (black circles), where multiple scattering resulted in deviations from traditional sigmoid (or S-shaped) curves from cuvette-based measurements. The deviations lead to failures when fitting the growth curve by sigmoid models (e.g., the Gompertz model) [24]. This issue could be potentially solved by well-designed, sophisticated calibrations using particles of different sizes at different concentrations [2]; however, such sophisticated calibrations require considerable efforts. Another possible solution is to make use of the time derivative of the OD curve, which has been attempted previously [25]; however, systematic studies remain missing, and it is unclear how the time derivative of the OD links to the bacterial growth parameters. Another important issue is that, in certain studies where the responses of microbes to reagents are investigated, the reagents of interest may contribute significantly to the scattering or absorption at the wavelength where the OD is measured and thus distort the OD curve. For example, adding polyvinylpyrrolidone (PVP)-coated cubic AgNPs to a bacterial culture not only vertically shifted the OD curve but also introduced elusive features, such as an additional peak, to the OD curve (Fig 1B). These issues might be partially resolved by using FL reporter in the microbes, assuming the reagents of interest do not interfere with the FL reporters. However, it remains unclear how the FL curves (Fig 1A, green squares) report the bacterial growth behaviors, such as the lag time and growth rate.

In this work, we developed a method for evaluating the growth of bacteria measured with multimode microplate readers. This method is based on the time derivatives of the OD and/or FL of the bacteria. Using quantitative models predicting the cell number and the number of fluorescent proteins as functions of time, we characterized the dependence of the first-order time derivative of OD and the second-order time derivative of FL on various parameters of the models, which are related to the commonly used lag time $\lambda$ and maximum specific growth rate $\mu$ in growth curves of microbes. The current method is consistent with traditional mathematical fittings of sigmoid growth curves; more importantly, it provides a better way for interpreting OD growth curves when the growth curve does not follow the well-established sigmoid shape due to multiple scattering. In addition, our method provides a framework for understanding the FL growth curves and for extracting the growth properties of bacteria from the FL measurements. The framework is especially useful when other components in the bacterial culture significantly contribute to the scattering and absorption (and therefore the OD). We demonstrated our method by applying it to the investigation of the lag time elongation observed in bacteria subjected to treatment with silver (Ag$^+$) ions. We observed that the results from our method corroborated very well with the traditional mathematical fittings of the OD curves. In

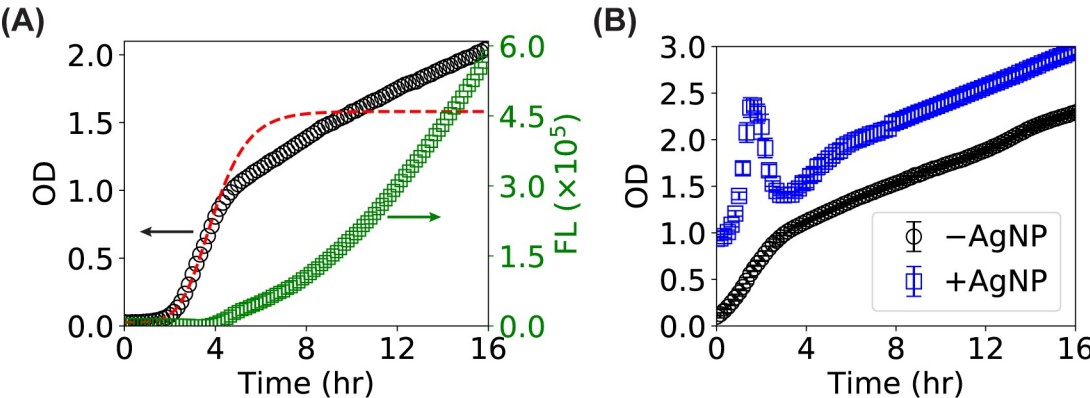

**Fig 1. Barriers for quantitative monitoring of the growth of bacteria using microplate readers.** (**A**) Representative curves of optical density (OD–black circles) and fluorescence (FL–green squares) of bacterial culture measured with a multimode 96-well microplate reader. Multiple scattering is significant in microplate-reader measurements, resulting in deviations of the OD growth curves from traditional cuvette-based measurements, leading to failures in fitting the growth curve by sigmoid models (such as the Gompertz model, red dashed line). The fluorescent growth curve of the same bacterial culture, which does not show the standard sigmoid shape, is much less studied. (**B**) Representative OD curves of bacteria measured with a multimode 96-well microplate reader in the absence (black circles) and presence (blue squares) of AgNPs. The AgNPs in the bacterial culture scatter light and interact with growth media and/or bacteria, causing not only a vertical shift in the OD measurements but also elusive features (e.g., a high peak at short time) in the OD growth curve. Error bars (smaller than the symbols) stand for the standard error of the mean.

addition, the current method was applied to the growth of bacteria in the presence of AgNPs at various concentrations, where the traditional growth measurements failed due to the high scattering, high absorption, and other interfering processes of the AgNPs. Our method allowed us to successfully extract the growth behavior of the bacteria from the FL measurements and understand how the growth was affected by the AgNPs.

## Materials and methods

### Models for the bacterial growth and expression of green fluorescent proteins

To predict the time derivatives of the cell number of and the number of fluorescent proteins in bacteria, we adopted simple models from the literature for the bacterial growth and GFP expression [26–28]. Following Juska et al. [27], we assume the bacteria in a dormant non-dividing state could be "activated" into an active dividing state (Fig 2A), for which we can write down the equations for the numbers of dormant bacteria ($n_D$) and active bacteria ($n_A$),

$$\frac{dn_D}{dt} = -\alpha \, n_D, \tag{1}$$

$$\frac{dn_A}{dt} = +\alpha \, n_D + k_0 \left(1 - \frac{n_A}{N}\right) n_A, \tag{2}$$

where $\alpha$ is the activation rate, $k_0$ is the maximum growth rate with unlimited nutrients, and $N$ is the maximum possible number of bacteria for a given limited amount of nutrients and resources. The total number of the bacteria, $n = n_D + n_A$, is related to the OD of the bacterial culture [4]. Following Leveau et al. [26], we model that the active bacteria grow and express green fluorescent proteins (GFP) with a generation rate of $g$ (Fig 2B) and that the newly expressed GFP are non-fluorescent as maturation (with a maturation rate of $k_m$) is needed for the GFP to be fluorescent (Fig 2B). Both the non-fluorescent ($GFP^n$) and fluorescent GFP

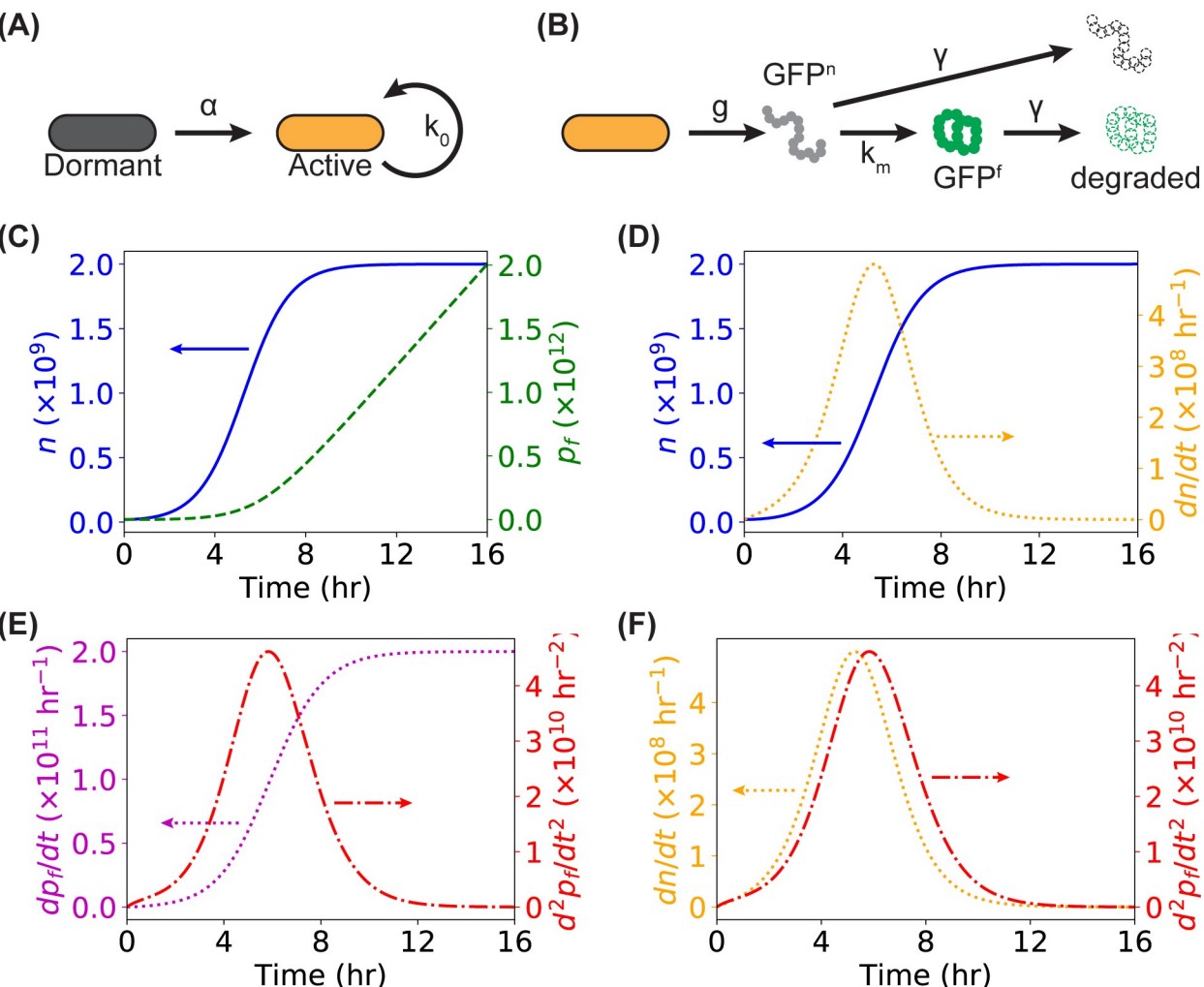

**Fig 2. Quantitative models and predictions for the cell number $n$ and the number of expressed and matured fluorescent proteins $p_f$. (A)** Sketch of the model for bacterial growth and proliferation: a bacterium recovers from a dormant state to an active reproduction state with an activation rate of $\alpha$, while activated bacteria grow and reproduce at a maximum growth rate of $k_0$. **(B)** Sketch of the model for the expression, maturation, and degradation of green fluorescent protein (GFP). Non-fluorescent GFP proteins (GFP$^n$), expressed in bacteria with a generation rate of $g$, will mature into fluorescent ones (GFP$^f$) with a maturation rate of $k_m$. Both GFP$^n$ and GFP$^f$ are degraded in bacteria, with a degradation rate of $\gamma$. **(C-F)** Predictions from the models in panels **A** and **B** (and Eqs 1 – 4) for the cell number ($n$), number of GFP$^f$ ($p_f$), and their time derivatives ($dn/dt$, $dp_f/dt$, and $d^2p_f/dt^2$) as functions of growth time.

(GFP$^f$) degrade with a degradation rate of $\gamma$ (Fig 2B). Therefore, we have the following equations for the numbers of non-fluorescent and fluorescent GFP proteins ($p_n$ and $p_f$, respectively):

$$\frac{dp_n}{dt} = gn_A - k_m p_n - \frac{\gamma p_n}{p_n + p_f + M} \tag{3}$$

$$\frac{dp_f}{dt} = k_m p_n - \frac{\gamma p_f}{p_n + p_f + M} \tag{4}$$

Note that we have assumed that the bacteria degrade both types of GFP molecules at the same rate and with the same degradation capacity $M$ [26]. It is also worthwhile to point out

that the number of bacteria ($n$, $n_D$, $n_A$, and $N$), the number of proteins ($p_n$ and $p_f$), and the degradation capacity ($M$) are dimensionless (i.e., unitless), while all the rates ($\alpha$, $k_0$, $g$, $k_m$, and $\gamma$) have the unit of 1/time (or, more specifically in this work, hr$^{-1}$).

We solved Eqs (1)–(4) numerically using the SciPy Python library [29], given the parameters of the models ($\alpha$, $k_0$, $N$, $g$, $k_m$, $\gamma$ and $M$) and the initial conditions ($n_D$, $n_A$, $p_n$, and $p_f$ at $t = 0$), and simulated the total cell number ($n$) and the number of GFP$^f$ ($p_f$) as functions of time ($t$). For simplicity, we assumed in the simulations that all the bacteria in a culture started with the dormant state, with an initial number of $0.05N$, and GFP proteins were not present initially, i.e., $n_D(0) = 0.05N$, $n_A(0) = 0$, $p_n(0) = 0$, and $p_f(0) = 0$. The time derivatives were then calculated numerically by $\frac{dn}{dt}\big|_{t_i} = \frac{n(t_{i+1}) - n(t_i)}{t_{i+1} - t_i}$, $\frac{dp_f}{dt}\big|_{t_i} = \frac{p_f(t_{i+1}) - p_f(t_i)}{t_{i+1} - t_i}$, and $\frac{d^2 p_f}{dt^2}\big|_{t_i} = \frac{\frac{dp_f}{dt}\big|_{t_{i+1}} - \frac{dp_f}{dt}\big|_{t_i}}{t_{i+1} - t_i}$.

## Preparation of Ag$^+$ solutions

Solutions of Ag$^+$ ions were prepared from AgNO$_3$ salt (Alfa Aesar), which was dissolved in autoclaved ultrapure water ($> 17.5$ MΩ) to reach a stock concentration of 100 mM [30]. The concentration of Ag$^+$ ions in the aqueous stock was measured and confirmed using inductively-coupled plasma mass spectrometry (Thermo Scientific iCAP Q ICP-MS) and a portable silver photometer (Hanna Instruments, MI). The stock solutions were aliquoted, shielded from light by aluminum foils, and stored at 4°C for later use within three weeks. The concentrations of Ag$^+$ ions in the aliquots was confirmed each time before use for experiments by the portable silver photometer.

## Synthesis of AgNPs

AgNPs were synthesized *via* polyol reduction method as described previously [30, 31]. Briefly, 50 mL of ethylene glycol (EG, J.T. Baker) was added to a 250-mL 3-neck round bottom flask and heated to 150°C in an oil bath, following by adding 0.6 mL of 3 mM NaHS (Alfa Aesar) in EG, 5 mL of 3 mM HCl (Alfa Aesar) in EG, 12 mL of 0.25 g polyvinylpyrrolidone (PVP, M$_W$ ~55,000, Sigma-Aldrich) in EG, and 4 mL of 282 mM CH$_3$COOAg (Alfa Aesar). The reaction proceeded at 150°C for ~1 h until the absorbance peak position of reaction mixture reached ~430 nm measured by UV-vis spectrometer. The reaction was then quenched by placing the flask in the ice bath. Acetone was added to the mixture at 5:1 volume ratio and the product was collected by centrifugation. The resultant PVP-coated cubic AgNPs were purified using water, collected by centrifugation, and re-suspended in water for future use.

## Bacterial strain and growth

An *E. coli* K-12 strain (MG1655) transformed with a high-copy number plasmid (with a pBR322 origin) encoding enhanced GFP (EGFP) and ampicillin resistance [7] was used in this study. The sequence map of the segment of the plasmid for the enhanced GFP and ampicillin is shown in S1 Fig in S1 File. The bacteria were grown at 37°C overnight in 6 mL Luria Broth (LB) medium supplemented with ampicillin in a shaking incubator at 250 rpm. On the second day, the overnight culture was diluted into fresh LB medium to reach OD = 0.05 (at 600 nm). Then Ag$^+$ ions or AgNPs were added to the fresh culture at desired final concentrations and mixed well before aliquoting to the 96-well clear bottom microplate for measurements with a microplate reader. The final concentration of Ag$^+$ ions was 40 μg/mL, while the final concentrations of AgNPs were 20, 40, 60, and 80 μg/mL. The fresh culture without adding Ag$^+$ ions or AgNPs (i.e., 0 μg/mL) was used as the negative control. Note that the enhanced GFP proteins were expressed continuously with a constitutive T7 promoter; therefore, induction was not needed nor used in this study.

### Bacterial growth curve measurements with a microplate reader

For the growth curve measurements with a microplate reader, 96-well clear bottom micro-plates (Corning Incorporated, NY) were first sterilized by incubating the wells with 200 proof ethanol for 5 mins. After pouring off the ethanol, the microplates were exposed to UV light at 254 nm for 15 mins. To avoid water condensation on the microplate lids during the measurements, the lids were coated with Triton X-100. Briefly, 4 mL of 0.05% Triton X-100 in 20% ethanol was added to each microplate lid and incubated at room temperature for 15 s, followed by pouring off the Triton solution [32]. The microplates were air dried before the measurements.

To measure the growth of bacteria, 200 μL of the prepared bacterial cultures (with or without Ag$^+$ ions or AgNPs) were transferred to the microplate wells. The microplates were covered with the pre-processed lids and placed in a microplate reader (BioTek Synergy H1M) to monitor the OD at 600 nm and the FL (excitation = 488 nm, emission = 525 nm) of the bacteria in the wells. Note that, to compensate the different beam path length other than the standard 1 cm, the OD values measured from the microplate reader were corrected by multiplying the raw values by 2.39, a pre-calibrated factor by a photometer (Implen Inc. CA). The plates were maintained at 37°C and rotated at 355 rpm. The OD and FL of each well were read every 10 mins for 16 hours. The mean and standard error of the mean (SEM) for each sample were calculated for each time point, followed by Hanning smoothing (with a window size of 11–15 data points) to reduce noises. Lastly, the time derivatives of the OD and FL were estimated numerically using $\frac{\Delta OD}{\Delta t}\big|_i = \frac{OD_{i+1}-OD_i}{\Delta t}$, $\frac{\Delta FL}{\Delta t}\big|_i = \frac{FL_{i+1}-FL_i}{\Delta t}$, and $\frac{\Delta^2 FL}{\Delta t^2}\big|_i = \frac{\frac{\Delta FL}{\Delta t}|_{i+1} - \frac{\Delta FL}{\Delta t}|_i}{\Delta t}$.

## Results

### Model-based prediction of time derivatives of bacterial growth and fluorescence

To predict the time derivatives of the cell number of and the number of fluorescent GFP in a bacterial culture, we adopted two simple models [26, 27] described in the "Methods and Materials" for the bacterial growth and GFP expression, as illustrated in Fig 2A and 2B. The bacterial growth model (Eqs 1 and 2) with three parameters (the activation rate $\alpha$, the maximum growth rate $k_0$, and the maximum possible bacterial number in a culture $N$) successfully produced the classical sigmoid growth curve (Fig 2C, blue solid line, with $\alpha = 1$ hr$^{-1}$, $k_0 = 1$ hr$^{-1}$, and $N = 2\times10^9$). In addition, the GFP expression model (Eqs 3 and 4) with four parameters (the generation rate $g$, the maturation rate $k_m$, the degradation rate $\gamma$, and the degradation capacity $M$) predicted that the fluorescence of a bacterial culture kept increasing without reaching plateaus (Fig 2C, green dashed line, with $g = 100$ hr$^{-1}$, $k_m = 1.5$ hr$^{-1}$, $\gamma = 500$ hr$^{-1}$, and $M = 2\times10^{11}$), resembling well with the experimental measurements from a microplate reader (Fig 1A, green squares). Note that, while reasonable values from the literature [7, 24, 26, 27] were used for the parameters in this example, detailed studies by varying the model parameters were performed in the sections below.

From the simulated curves of $n(t)$ and $p_f(t)$, we then examined their time derivatives numerically. The first-order time derivative of the cell number ($dn/dt$) showed a bell-shaped peak (Fig 2D, orange dotted line), while the first-order time derivative of the number of GFP$^f$ ($dp_f/dt$) showed a sigmoid shape (Fig 2E, purple dotted line), similar to the curve of $n(t)$ (Fig 2C, blue solid line) and suggesting the possibility of using the fluorescence to report the growth of the bacteria. In addition, the second-order time derivative ($d^2 p_f/dt^2$) showed a bell-shaped peak (Fig 2E, red dash-dotted line) and appeared very similar to $dn/dt$. We note that there is a small horizontal shift between the two peaks (Fig 2F), which was further investigated in more details as shown below.

## Dependence of time derivatives on the activation rate $\alpha$

As the activation rate describes how quickly the dormant bacteria transit into the growing, dividing, active ones, it influences the time a bacterial culture takes to adjust to the environment, and thus determines the lag time of the bacterial growth [27]. Therefore, it is expected that a higher activation rate results in shorter lag time and thus a lower location of the peaks of the time derivatives (i.e., $dn/dt$ and $d^2p_f/dt^2$). To quantitatively estimate the dependence of the time derivatives on the activation rate $\alpha$, we solved the Eqs (1)–(4) and performed numerical simulations with varying $\alpha$ from $10^{-4}$ to $10^2$ hr$^{-1}$, while keeping the other parameters constant ($k_0 = 1$ hr$^{-1}$, $N = 2\times10^9$, $g = 100$ hr$^{-1}$, $k_m = 1.5$ hr$^{-1}$, $\gamma = 500$ hr$^{-1}$, and $M = 2\times10^{11}$). As expected, the cell-number curve shifted to the left as the activation rate increased (Fig 3A), while the fluorescence growth curve showed a similar trend (Fig 3B). The simulated results showed bell-shaped peaks in the time derivatives (Fig 3C and 3D). We found that increasing the activation rate led to horizontal shift of the peaks to the left while the peak heights did not change. We quantitatively determined the peak locations (i.e., the time points corresponding to the maxima of the peaks, $\tau_p$ and $\tau_p^f$, respectively) and observed that both $\tau_p$ and $\tau_p^f$ exponentially decreased as the activation rate increased for $\alpha \leq 1$ hr$^{-1}$ (Fig 3E, black circles and green triangles), while the peak locations were much less sensitive to higher $\alpha$ ($\geq 1$). In addition, we observed that the difference between $\tau_p^f$ and $\tau_p$ (i.e., $\tau_p^f - \tau_p$) remained constant with varying $\alpha$ (Fig 3E and S2A Fig in S1 File), indicating that the activation of the bacteria from dormancy to the active growing state did not play a role in the horizontal shift between the OD-based peak and fluorescence-based peak observed in Fig 2F.

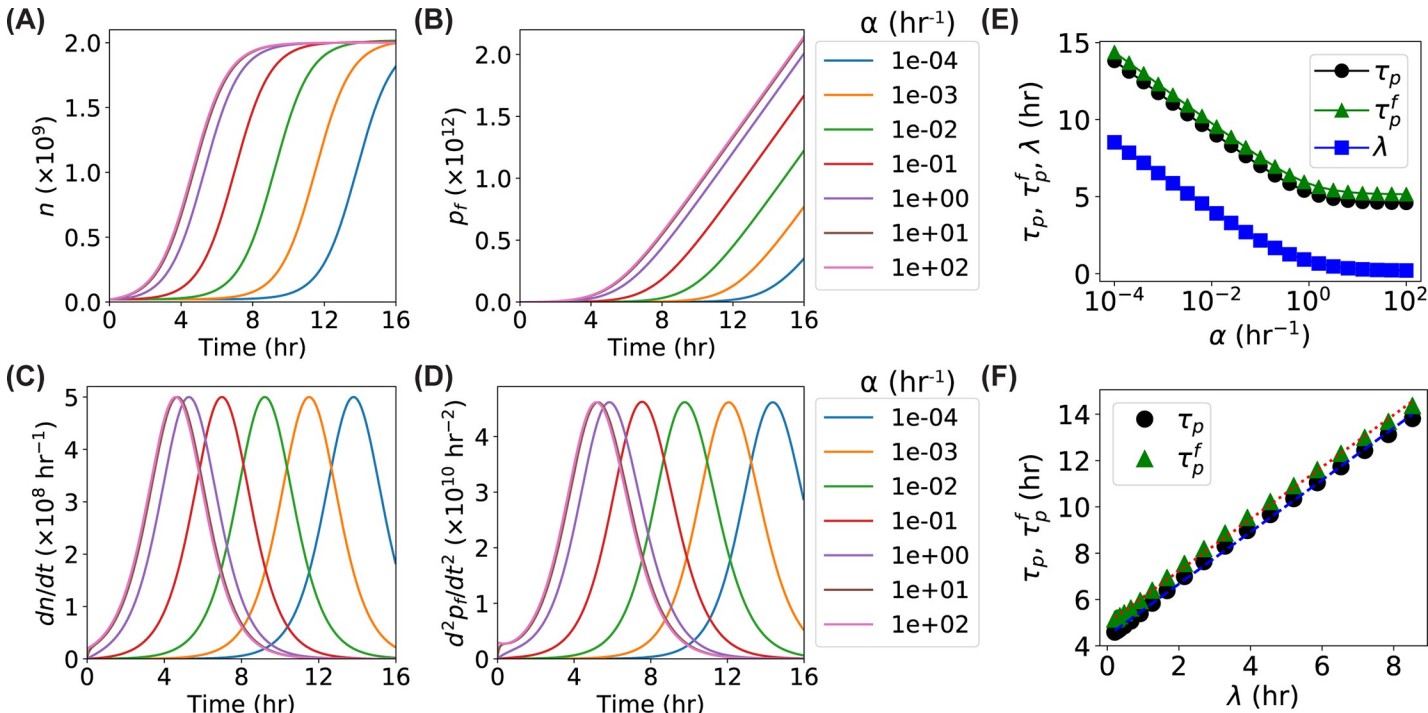

**Fig 3. Predictions of the models for the dependence of the cell number $n$ the number of GFP$^f$ $p_f$, and their time derivatives on the activation rate $\alpha$. (A-D)** Predicted curves of the models for the (A) $n$, (B) $p_f$, (C) $dn/dt$, and (D) $d^2p_f/dt^2$ as functions of time with increasing activation rates $\alpha$ ranging from $10^{-4}$ to $10^{+2}$ hr$^{-1}$. (**E**) Predicted dependence of the peak locations ($\tau_p$–black circles and $\tau_p^f$–green triangles) of the $dn/dt$ and $d^2p_f/dt^2$ curves on the activation rate $\alpha$, compared to that of the fitted lag time $\lambda$ (blue squares). (**F**) Relation between the peak locations ($\tau_p$–black circles and $\tau_p^f$–green triangles) and the fitted lag time $\lambda$. Blue dashed line and red dotted line are linear fittings.

To further confirm that the peak locations ($\tau_p$ and $\tau_p^f$) were capable of reporting the lag time ($\lambda$) of the bacterial culture, we fitted the logarithm of the simulated $n(t)$ curves by the Gompertz model, $\log\left(\frac{n(t)}{n(0)}\right) = A \exp\left\{-\exp\left[\frac{\mu \cdot e}{A}(\lambda - t) + 1\right]\right\}$, where $\mu$ is the maximum specific growth rate and $\lambda$ is the lag time [19, 24]. Comparing the fitted lag times (Fig 3E, blue squares) with the peak locations showed that the dependence of the lag time $\lambda$ on the activation rate was the same as the peak locations, although the vertical baselines were different. We also observed that both the peak locations were linear to the lag time (Fig 3F) with the same slope that is close to one ($1.12 \pm 0.01$), suggesting that the peak locations could be used to report the lag time of the bacterial growth, although a slight inflation exists (as the slope is $1.12 > 1$).

### Dependence of time derivatives on the maximum growth rate $k_0$

The maximum growth rate $k_0$ in the model describes how fast the bacteria divide in the presence of unlimited nutrients and resources [27]; therefore, it is expected that $k_0$ affects the heights of the peaks in the time derivatives. We ran simulations by varying $k_0$ from 0.5 to 2.0 hr$^{-1}$, while keeping the other parameters constant ($\alpha = 1$ hr$^{-1}$, $N = 2\times10^9$, $g = 100$ hr$^{-1}$, $k_m = 1.5$ hr$^{-1}$, $\gamma = 500$ hr$^{-1}$, and $M = 2\times10^{11}$). As expected, $n(t)$ became steeper as the growth rate $k_0$ increased (Fig 4A), resulting in higher peaks in $dn/dt$ (Fig 4C). We also observed that the peak locations shifted to the left at higher $k_0$ (Fig 4C). Similar effects were observed for $d^2p_f/dt^2$ (Fig 4B and 4D). The peak height ($\eta_p$) of $dn/dt$ was linear to the growth rate $k_0$ (Fig 4E, black circles), similar to the $\mu-k_0$ relation where $\mu$ is the maximum specific growth rate from the

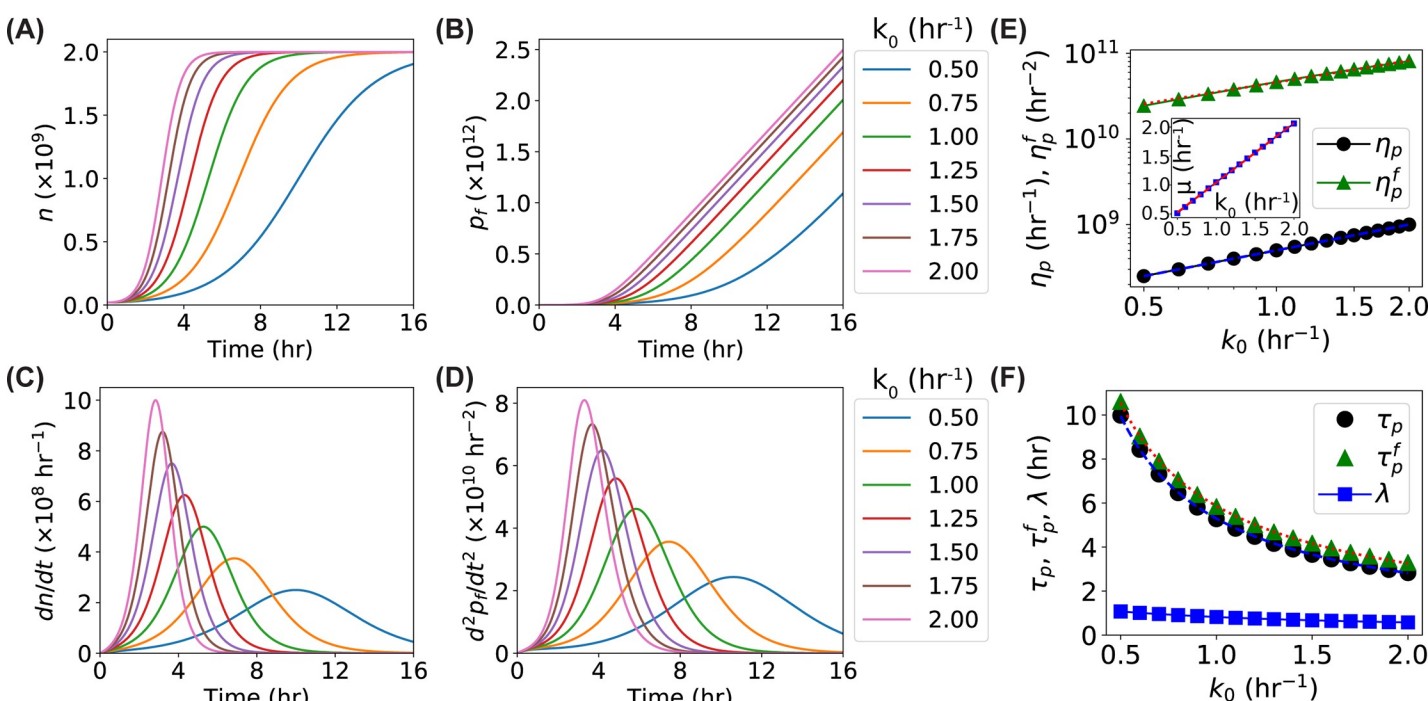

**Fig 4. Predictions of the models for the dependence of the cell number $n$, the number of GFP$^f$ $p_f$ and their time derivatives on the maximum growth rate $k_0$.**
(A-D) Predicted curves of the models for the (A) $n$, (B) $p_f$, (C) $dn/dt$, and (D) $d^2p_f/dt^2$ as functions of time with increasing growth rates $k_0$ ranging from 0.50 to 2.00 hr$^{-1}$. (E) Predicted dependence of the peak heights ($\eta_p$–black circles and $\eta_p^f$–green triangles) of the $dn/dt$ and $d^2p_f/dt^2$ curves on the growth rate $k_0$ in the model. Inset: predicted dependence of the fitted maximum specific growth rate $\mu$ on the growth rate $k_0$ in the model. (F) Predicted dependence of the peak locations ($\tau_p$–black circles and $\tau_p^f$–green triangles) of the $dn/dt$ and $d^2p_f/dt^2$ curves and the fitted lag time ($\lambda$ –blue squares) on the growth rate $k_0$. Blue dashed line and red dotted line are exponential fittings.

Gompertz fitting (as observed in the inset of Fig 4E). However, the dependence of the peak height ($\eta_p^f$) of $d^2p_f/dt^2$ deviated significantly from a line (Fig 4E, green triangles). Interestingly, the power law, $\eta_p^f = a_1 k_0^c$, (i.e., linear in a log-log scale, $\log \eta_p^f = c \log k_0 + \log a_1$) was more appropriate. Fitting the data gave an exponent of 0.84 (Fig 4E, red dotted line). It is noted that performing the power law fitting on $\eta_p$ resulted in an exponent of 1.00, confirming the linearity between $\eta_p$ and $k_0$ (Fig 4E, blue solid line). We also observed that the relation between the peak locations ($\tau_p$ and $\tau_p^f$) and the growth rate $k_0$ followed the power law with exponents of -0.91 and -0.85, respectively (Fig 4F, black circles and green triangles). Note that the observed large changes of the peak locations (~7 hr from $k_0 = 0.5$ to $2.0$ hr$^{-1}$) did not necessarily correspond to large changes in the lag time $\lambda$. In contrast, fitting $\log\left(\frac{n}{n_0}\right)$ with the Gompertz model [19, 24] showed a much smaller change ($\leq 0.5$ hr) in the lag time for the same $k_0$ range (Fig 4F, blue squares). We also observed that the difference between $\tau_p^f$ and $\tau_p$ showed slight dependence on the maximum growth rate $k_0$ (S2B Fig in S1 File).

## Dependence of time derivatives on the GFP expression rate $g$ and maturation rate $k_m$

We then examined how the time derivatives depend on the GFP expression rate $g$ and maturation rate $k_m$. Since these rates are not related to the activation and division of bacteria, the cell number does not depend on these two parameters (see Eqs 1 and 2). Therefore, $dn/dt$ is independent on $g$ and $k_m$. On the other hand, we expected that these two rates significantly affect the number of fluorescent GFP $p_f$ in the bacterial culture [26] and thus its time derivatives. To test this hypothesis, we ran simulations with either varying the expression rate or the maturation rate (one at a time) while keeping the other parameters constant. As shown in Fig 5A, $p_f$ became much steeper as the expression rate $g$ increased, resulting in higher peaks in the second-order time derivative $d^2p_f/dt^2$ (Fig 5A, inset). However, the peak location was not affected (Fig 5A, inset); therefore, $\tau_p^f - \tau_p$ was independent on the expression rate $g$ (Fig 5B and S2D Fig in S1 File). More quantitatively, we found the peak height $\eta_p^f$ was linearly dependent on the expression rate when all the other parameters were kept constant (Fig 5B). When varying the maturation rate $k_m$, we observed that not only the peak heights but also the peak locations were affected. As shown in Fig 5C, $p_f$ became steeper at higher maturation rates, resulting in higher peaks in the time derivatives (Fig 5C, inset). In addition, the peaks were shifted to the left as the maturation rate increased (Fig 5C, inset). Quantifying the peak location and height showed that the dependence of the peak location $\tau_p^f$ on the maturation rate $k_m$ followed a power law, with an exponent of -0.11 (Fig 5D, blue triangles). As a result, the difference between $\tau_p^f$ and $\tau_p$ was significantly dependent on the maturation rate $k_m$ (Fig 5D and S2E Fig in S1 File), suggesting that the horizontal shift between the fluorescence-based peak and the OD-based peak observed in Fig 2F was largely due to the maturation of the fluorescent proteins. Interestingly, the relation between the peak height $\eta_p^f$ and the maturation rate $k_m$ resembled the Michaelis–Menten kinetics [33] curve (Fig 5D, black circles), although the underlying reason is unclear. The simulated curve could be fitted well with $\eta_p^f = \frac{V_M k_m}{k_m + K_M} + b$, giving $K_M = 0.26$ hr$^{-1}$.

## Dependence of time derivatives on the GFP degradation

We lastly investigated how the second-order time derivative of the fluorescence curve depends on the GFP degradation by running simulations with either varying degradation rate $\gamma$ or

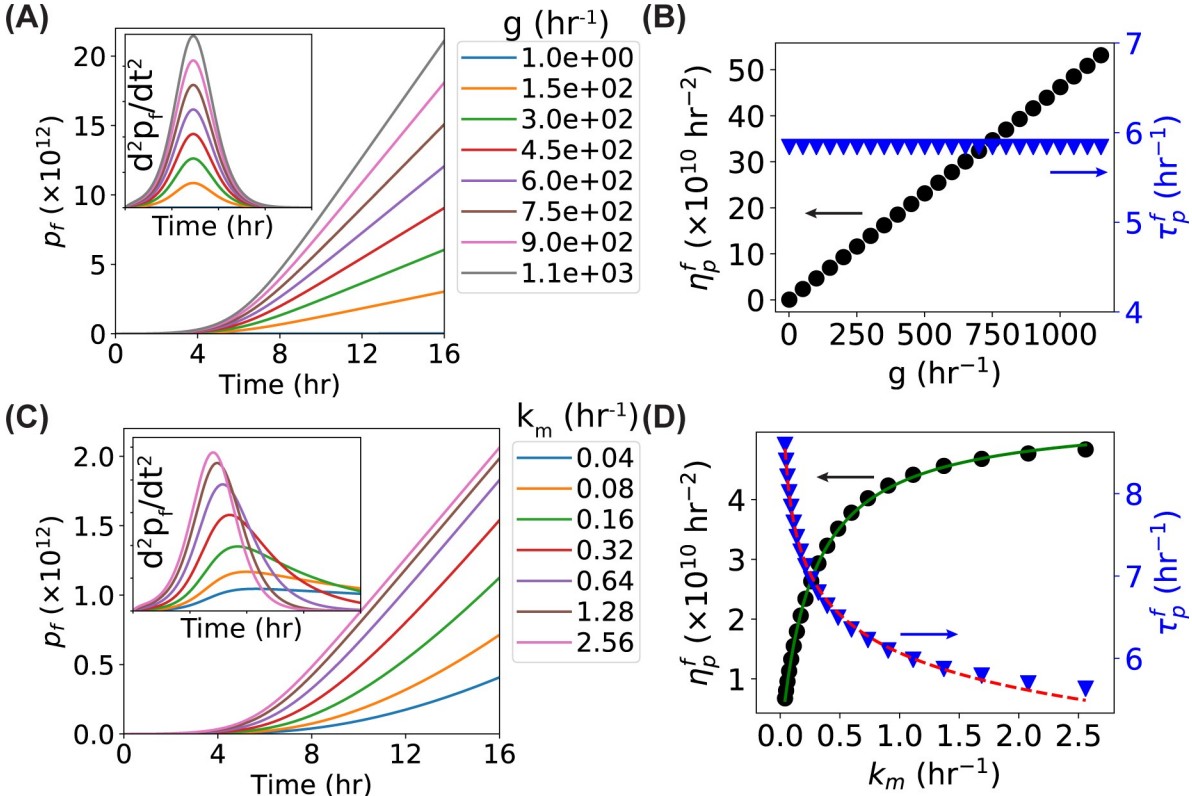

**Fig 5. Predictions of the models for the dependence of the cell number *n*, the number of GFP$^f$ $p_f$, and their time derivatives on the GFP generation rate *g* and maturation rate $k_m$.** (A) Predicted curves of the models for $p_f$ and $d^2p_f/dt^2$ (inset) as functions of time with increasing generation rate *g* ranging from 1 to $1.1\times10^3$ hr$^{-1}$. (B) Predicted dependence of the peak height ($\eta_p^f$–black circles) and peak location ($\tau_p^f$–blue triangles) of the $d^2p_f/dt^2$ curve on the GFP generation rate *g*. (C) Predicted curves of the models for $p_f$ and $d^2p_f/dt^2$ (inset) as functions of time with increasing maturation rate $k_m$ ranging from 0.04 to 2.56 hr$^{-1}$. (D) Predicted dependence of the peak height ($\eta_p^f$–black circles) and peak location ($\tau_p^f$–blue triangles) of the $d^2p_f/dt^2$ curve on the GFP maturation rate $k_m$. Green solid line and red dashed line are fittings.

varying degradation capacity *M* (one at a time) while keeping all the other parameters constant. We observed that the number of fluorescent GFP $p_f$ became shallower as the degradation rate $\gamma$ increased, resulting in a decrease in the peak height ($\eta_p^f$) of $d^2p_f/dt^2$ (Fig 6A). More interestingly, at high enough degradation rate, the $p_f(t)$ curve started to show slower growth at longer times or even plateaus (e.g., $4\times10^{11}$ hr$^{-1}$) (Fig 6A). This change in the growth curve was reflected by the dip after the peak in $d^2p_f/dt^2$ and a slight left shift in the peak location (Fig 6A, inset). Quantifying the peak height and location showed that both of the peak height and peak location were almost constant below $\gamma<10^{10}$ hr$^{-1}$ (Fig 6B inset, with $M = 2\times10^{11}$) but decreased quickly at higher degradation rates (Fig 6B). For the degradation capacity (*M*), we observed higher *M* values in general gave steeper $p_f$ curves (Fig 6C), which is expected as a larger degradation capacity (but the same degradation rate $\gamma$) indicates that more GFP proteins are degraded per unit time [26]. Interestingly, a more careful examination showed that the dependencies of both peak height and peak location on the degradation capacity were more complicated than monotonic changes (Fig 6C, inset). Quantifying the peak locations and heights showed that interesting dependence on the degradation capacity (Fig 6D). We also note that both the degradation rate and capacity affected $\tau_p^f - \tau_p$, the horizontal shift between the fluorescence-based peak and OD-based peak (Fig 2F).

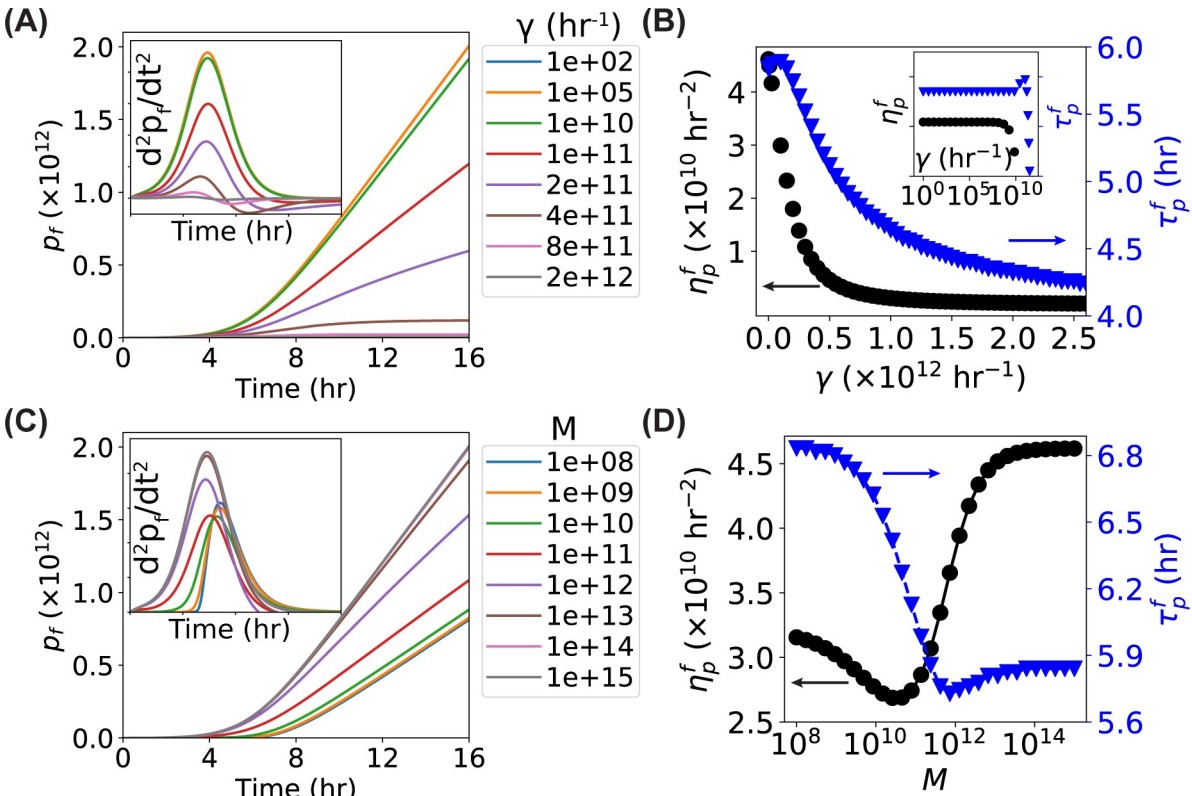

**Fig 6. Predictions of the models for the dependence of the cell number *n*, the number of GFP$^f$ $p_f$, and their time derivatives on the GFP degradation rate *γ* and degradation capacity *M*.** (A) Predicted curves of the models for $p_f$ and $d^2p_f/dt^2$ (inset) as functions of time with increasing degradation rate *γ* ranging from 100 to $2\times10^{12}$ hr$^{-1}$. (B) Predicted dependence of the peak height ($\eta_p^f$–black circles) and peak location ($\tau_p^f$–blue triangles) of the $d^2p_f/dt^2$ curve on the GFP degradation rate *γ*. Inset: a close-up look of the same data in the range of $\gamma\in[1, 10^{10}]$ hr$^{-1}$. (C) Predicted curves of the models for $p_f$ and $d^2p_f/dt^2$ (inset) as functions of time with increasing degradation capacity *M* ranging from $10^8$ to $10^{15}$. (D) Predicted dependence of the peak height ($\eta_p^f$–black circles) and peak location ($\tau_p^f$–blue triangles) of the $d^2p_f/dt^2$ curve on the GFP degradation capacity *M*.

## Application of the time derivative based method to study lag time elongation caused by Ag⁺-treatment

We have been investigating the antimicrobial activity and mechanism of silver (Ag) in various forms, such as ions and nanoparticles [7, 30, 34, 35]. For example, we performed growth-curve measurements on *E. coli* bacteria in the presence of Ag⁺ ions following standard protocols (i.e., using cuvettes and photometry), and found that the major effect of Ag⁺ ions was to elongate the lag time [7]. Here we repeated the growth experiments of GFP-expressing *E. coli* bacteria in the absence and presence of Ag⁺ ions at 40 µM with a multimode microplate reader by measuring both the optical density at 600 nm (OD) and green fluorescence (FL). We observed that, due to significant multiple scattering, the OD curve from the microplate reader did not follow the sigmoid shape (Fig 7A), even if the overnight culture (i.e., 16 hr) was expected to be in the stationary phase. This observation was consistent with previous results reported in the literature [2]. On the other hand, the growth curves at low OD values (e.g., ≤ 0.8–1.0) confirmed that the presence of Ag⁺ ions elongated the lag time while keeping the growth rate (i.e., the slope) roughly the same (Fig 7A), reproducing our previous results [7]. More quantitatively, we fitted the logarithm of the OD curves partially (using data with OD ≤ 1.0) by the Gompertz model [24] and found that the lag times were 1.38±0.02 hr and 6.6±0.9 hr,

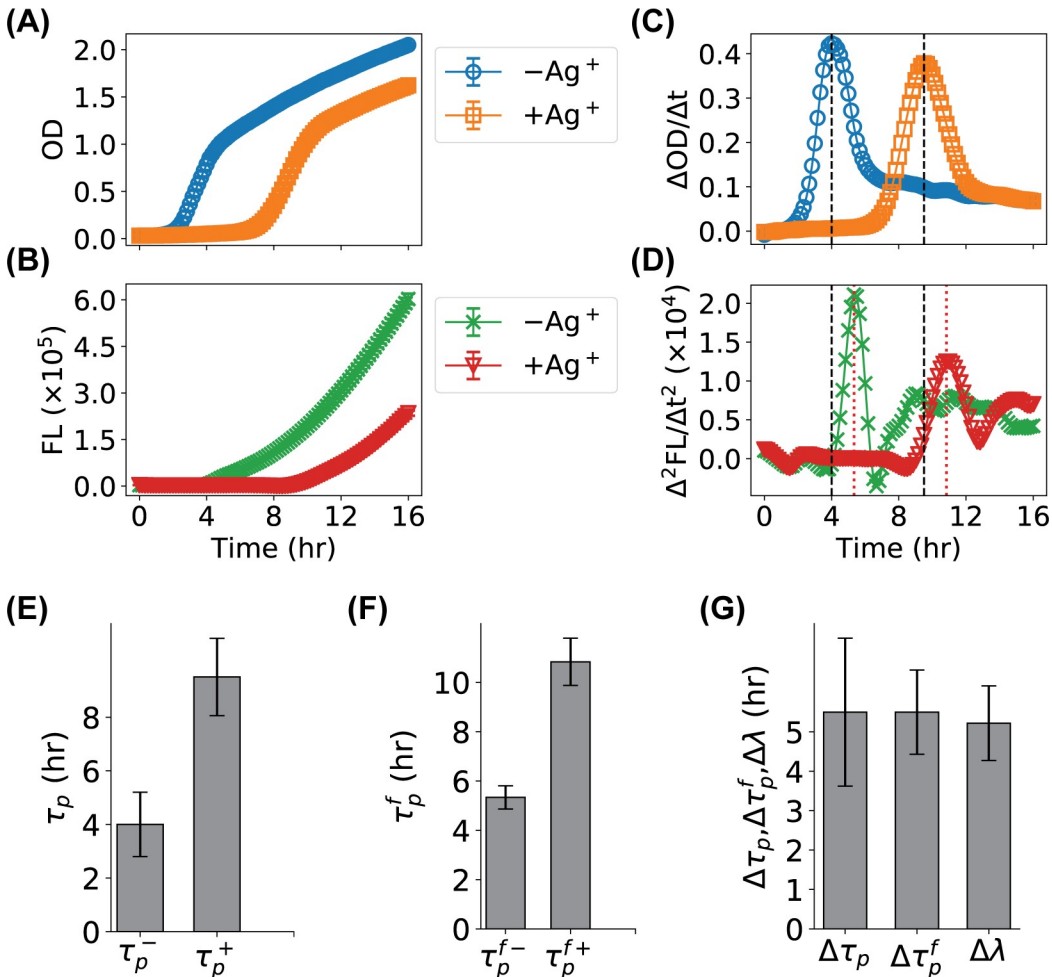

**Fig 7. Application of the time-derivative based method for identifying the elongation of lag time in the growth of *E. coli* bacteria due to the treatment with Ag$^+$ ions.** (**A**) OD growth curves of *E. coli* in the absence (-Ag$^+$, blue circles) and presence (+Ag$^+$, orange squares) of 40 μM Ag$^+$ ions, measured with a multimode 96-well microplate reader. (**B**) FL growth curves of the same *E. coli* samples (-Ag$^+$: green crosses; Ag$^+$: red triangles). Error bars in panels **A** and **B** represent standard errors of the means (SEM). (**C, D**) Time derivatives of the corresponding growth curves in panels **A** and **B**. Vertical lines highlights the corresponding peaks. (**E**) Measured peak locations of ΔOD/Δt in the absence ($\tau_p^-$) and presence ($\tau_p^+$) of Ag$^+$ ions. Error bars stand for the standard deviations of the peaks. (**F**) Measured peak locations of Δ$^2$FL/Δt$^2$ in the absence ($\tau_p^{f-}$) and presence ($\tau_p^{f+}$) of Ag$^+$ ions. Error bars stand for the standard deviations of the peaks. (**G**) Comparison between the changes of the peak locations ($\Delta\tau_p$ and $\Delta\tau_p^f$) and the elongation of the fitted lag time ($\Delta\lambda$).

respectively, corresponding to a lag time elongation of 5.2±0.9 hr. The reported errors here were fitting errors, which were large in some cases (e.g., +Ag$^+$ ions) presumably due to the missing of the bacterial growth data above 1.0.

We calculated the first-order time derivatives of the OD curves (ΔOD/Δt), which showed distinct peaks (Fig 7C). In the absence of Ag$^+$ ions, the peak was centered at 4 hr with a standard deviation (STDEV) of 1.2 hr (Fig 7C, blue circles). In contrast, the peak for the Ag$^+$-treated bacteria was centered at 9.5 hr with a STDEV of 1.4 hr (Fig 7C, orange squares). The shift in the peak location due to the Ag$^+$-treatment was 5.5 ± 1.9 hr (Mean ± STDEV, Fig 7E), consistent with the lag time elongation estimated from the Gompertz fitting (Fig 7G).

We also measured the fluorescence curves of the same samples in the absence and presence of Ag$^+$ ions (Fig 7B). The shapes of the measured FL curves were similar to the predictions

from the models (Figs 2–6), providing evidence to support the validity of the models used in this work. More importantly, we calculated the second-order time derivative ($\Delta^2FL/\Delta t^2$) and observed the model-predicted peaks centered at 5.3 hr and 10.8 hr, respectively (Fig 7D). The shift in the peak location was 5.5 ± 1.1 hr (Mean ± STDEV, Fig 7F), corroborating very well with the shift from the OD results and the lag time elongation from the Gompertz fitting (Fig 7G).

## Applying the time derivative based method to study the growth *E. coli* bacteria in the presence of AgNPs

When attempting to measure the OD growth curves of bacteria treated with PVP-coated cubic silver nanoparticles (AgNPs) at various concentrations (0–80 µg/mL) using the microplate reader, we observed that the OD curves were significantly affected by the presence of the AgNPs (Fig 8A, inset). First, we observed a shift in the baseline of the OD measurements, presumably due to the contributions of the AgNPs to the absorption and scattering of light. More importantly, strange peaks emerged in the OD curves at short time, possibly due to the aggregation, dissolution, and other processes of the AgNPs in the growth media. These peaks in the OD curves made it challenging to directly apply the Gompertz model to quantitatively obtain the growth properties of the bacteria, without making various assumptions about the peaks.

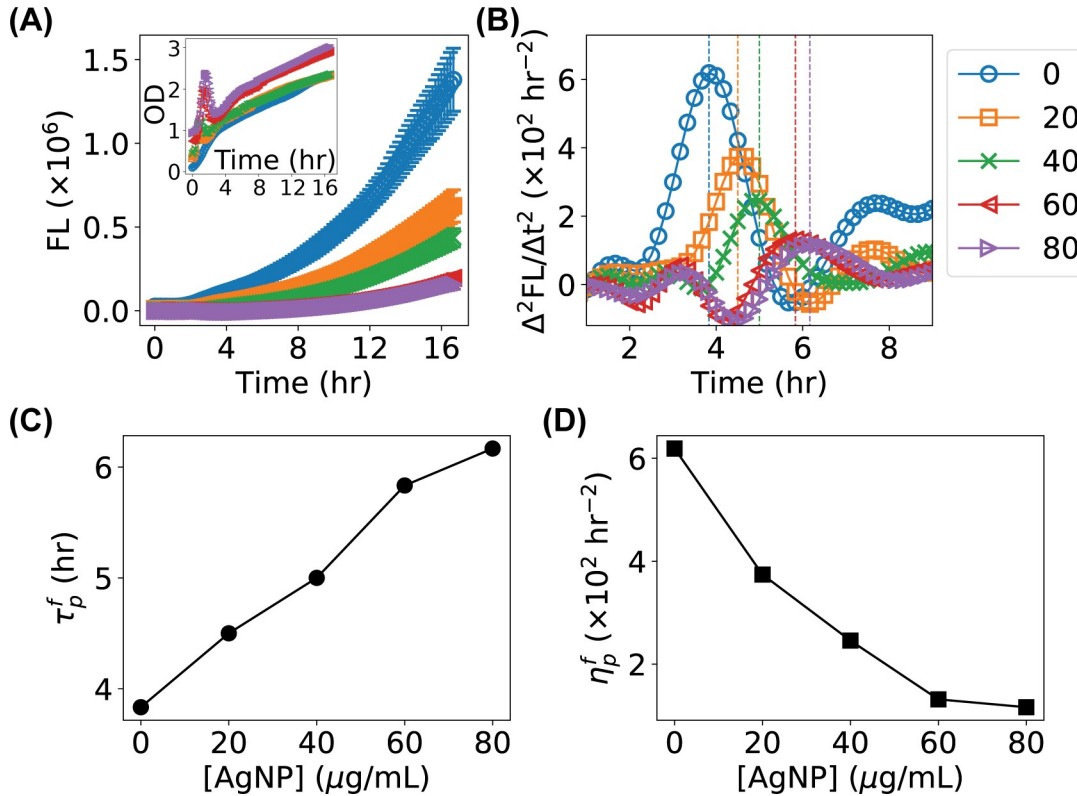

**Fig 8. Application of the time-derivative based method for identifying the effect of silver nanoparticles (AgNPs) at various concentrations on the growth of *E. coli* bacteria.** (**A**) FL growth curves of *E. coli* bacteria in the presence of AgNPs at 0 (control, blue circles), 20 (orange squares), 40 (green cross), 60 (red triangle), and 80 (purple triangle) µg/mL. Inset: the corresponding OD growth curves for the same samples. Error bars represent standard errors of the means (SEM). (**B**) Time derivatives of the FL growth curves ($\Delta^2FL/\Delta t^2$) in panel **A**. Vertical lines highlight the peak locations. (**C, D**) Measured dependence of the (**C**) peak locations $\tau_p^f$ and (**D**) peak heights $\eta_p^f$ on the concentration of AgNPs.

On the other hand, we found that the fluorescence curves followed the predictions from the models in the presence of the AgNPs (Fig 8A). We applied the time derivative based method on the FL curves and observed that the peak centers in $\Delta^2 FL/\Delta t^2$ were shifted to the right at higher concentrations of AgNPs, along with decreases in the peak heights (Fig 8B). It is noted that Hanning smoothing of curves was performed to reduce noises here. Characterizing the peaks showed that the peak location $\tau_p^f$ increased linearly, and the peak height $\eta_p^f$ decreased exponentially, as the concentration of AgNPs increased.

## Discussion

In this study, we used a gram-negative bacterium, *E. coli*, at 37°C as a model organism. However, we expect that the current method is readily applicable to other microorganisms such as gram-positive bacteria or yeast at different temperatures, because the current method does not rely on the biological properties of the microorganisms and OD-based growth curves have been commonly and widely used for different bacteria and other microorganisms, which show similar sigmoid growth curves.

It is worthwhile to note that the major purpose of developing the current method is not to replace the traditional OD measurements and mathematical fittings (e.g., Gompertz model) of growth curves. Instead, it is a complementary means to assess the growth behaviors of microbes especially when the traditional OD curves deviate from the sigmoid shape due to multiple scattering or contributions from other reagents of interest in the studies. It is also important to point out several caveats and limitations in the current method. First, our method relies on the peaks in the time derivatives of the OD-based or fluorescence-based growth curves, which require implicitly the existence of maximum rate in the growth of the microorganism. However, this requirement is not necessarily met for some microorganisms or at certain temperatures. Second, taking the time derivatives of the growth curves generally introduces additional numerical errors, which possibly demolish or distort the peaks and thus lead to failure of the current method. We also note that, as the current method relies on the OD and fluorescence measurements, its sensitivity and robustness at low bacterial concentrations is limited by the sensitivity of the instrument for OD and fluorescence quantifications.

It is also important to clarify that we highly recommend using the time derivatives of both the OD and fluorescence growth curves when possible (as we did in Fig 7). The peak properties of the fluorescence-based growth curves provide additional, useful information for quantitatively evaluating the growth of microbes. However, the time derivatives from the fluorescence curves are not sufficient for a complete description of the bacterial growth.

A horizontal shift was observed between the peak based on fluorescence-based growth curve and the peak based on the corresponding OD-based growth curve (Figs 2F and 7D). This shift is mainly due to the maturation and degradation of the fluorescent proteins. A systematic comparison showed that this horizontal shift depends mostly on the maturation rate $k_m$, significantly on the degradation rate $\gamma$ and degradation capacity $M$, slightly on the maximum growth rate $k_0$, but not on the activation rate $\alpha$, maximum cell number $N$, and fluorescent protein expression rate $g$ (S2 Fig in S1 File).

The dependencies of the changes in the peak location (horizontal shift) and peak height (vertical shift) on the various growth parameters ($\alpha, k_0, N, g, k_m, \gamma, M$) are many-to-many mapping. In other words, a change in the peak location or height can possibly depend on several parameters, and one parameter can possibly affect both the location and height of the peaks. On one hand, this many-to-many relationship brings complications if it is desired to compare the growth rate (related to $k_0$) and lag time (related to $\alpha$) from the fluorescence-based peaks among samples with different fluorescence-related parameters. On the other hand, the many-

to-many mapping could be an advantage in many scenarios, for example, where toxins or reagents cause different responses in the OD-based and fluorescent-based growth curves. One of such examples is that polydopamine causes significant changes in the fluorescent-based growth curves but has much less effects on the OD-based growth curves [36, 37].

In addition to the growth rate ($\mu$) and lag time ($\lambda$), another commonly used quantity for describing bacterial growth is the mean generation time or doubling time ($T_d$). Because $T_d$ is related to the maximum growth rate, $T_d \propto 1/k_0$ [27], and the peak height $\eta_p$ from the OD-based growth curve is linearly dependent on and only on $k_0$ (Fig 4E), the mean generation time can be readily reported by the current method using the peak height of the time derivative of the OD growth curve. On the other hand, if the OD-based growth curves are significantly distorted by presence of other reagents and only the fluorescence-based growth curves are available (e.g., Fig 8A), it is practically difficult to quantify the mean generation time because of the many-to-many relationship between the peak properties of the time derivatives and the various parameters.

We chose to use and examine the cell number $n(t)$, instead of the OD, in all our simulated results. This choice was made because the relation between the cell number (or concentration) and the OD (or absorbance/transmittance) is complicated when taking into account multiple scattering and instrument parameters (e.g., aperture shape, aperture radius, and the distance between detector and sample) [38]. In certain regimes, a parabolic dependence is valid for the relation of OD *vs.* cell number [2, 38]. However, this dependency is invalid in general, requiring "ideal instrumentation and nonabsorbing bacteria" [38]. As it was out of the scope of the current study, we did not convert cell numbers to OD values in the simulations for reproducing the flattening/saturation effects of multiple scattering on the OD growth curves (Figs 1A and 7A). However, it is worthwhile to discuss why taking the first derivative of the OD growth curve helps to solve the problem of multiple scattering (or nonlinear OD-concentration relation). The reason is that the flattening/saturation of the OD growth curve caused by multiple scattering gives lower/shallower slopes, which in turn lead to lower values for the first derivative of the curve (as the first derivative of a function/curve is the slope of the tangent line to the curve). As a result, the regimes affected by multiple scattering would appear as shoulder(s) or long tail(s) to the right of the peak, as shown in Fig 7C. The shoulder, or long tail, does not come into play when identifying the peak, and thus the peak location and height are not affected (if the shoulder is not too significant). Therefore, the effect of the multiple scattering on the OD curve is lessened or possibly removed by taking the first derivative of the OD growth curve.

The current work offered a method for analyzing the OD and fluorescence growth curves from microplate readers *without* calibrations for eliminating the effects of multiple scattering. However, the current method can be combined with growth curves generated with careful calibrations [2, 39]. Recently, a beautiful interlaboratory study was carried out, concluding that serial dilution of silica microspheres worked the best for easy, accurate, and robust calibration of an instrument for converting OD to cell number [39], which is expected to be very useful in the field of microbiology. We believe that these accurate and robust calibration methods will provide better data for the current method to start with.

An advantage of our method is that the analysis based on the time derivatives of the OD and FL growth curves could be non-parametric and model-free. Although we have used models with several parameters to help us to understand the time derivatives in this work, the application of our method does not require any model. In our method, one only needs to calculate the derivatives of the measured time series of OD and FL, locate the peaks in the time derivatives, and determine the peak locations and peak heights. The changes in the peak locations and peak heights then report how the growth properties of the bacteria are affected by the

reagents of interest. This analysis does not require mathematical equations and fitting; it works without any underlying assumptions on the microbial growth. We expect that the non-parametric and model-free nature makes our method robust and easy to use when monitoring microbial growth.

Furthermore, it is possible to connect the non-parametric changes in the peaks with parametric models in order to understand the mechanism of the reagents of interest to affect the growth of bacteria. In this work, we adopted very simple models for the growth of bacteria and expression of fluorescent proteins. However, there are better and more sophisticated models available in the literature [7, 8, 40], and new models can be developed if desired. For example, if plasmid loss is significant in certain cases (and thus the expression of fluorescent proteins is affected), it might be desired to modify the model for the expression of fluorescent proteins. If cell death is of interest (e.g., for studies of the effects of toxins and/or toxicants at high concentrations), it would be wanted to include the cell death in the models (e.g., S3 Fig in S1 File). It might also be necessary to include coupled model parameters in some situations, for example, where the expression and fluorescence of fluorescent proteins are linked to the growth of bacteria. Although the current method does not require a model, it is expected to be extendable, versatile, and compatible with different, sophisticated models for different applications.

The current method is based on OD and fluorescence with microplate readers and thus reports the growth properties of bacteria at the ensemble level (i.e., the average over many bacteria). Therefore, the current method cannot differentiate live and dead cells, which is a limitation compared to time-lapse microscopy that has been used to monitor the growth and morphology of microbes in real-time [35, 41]. For example, we previously investigated the growth of bacteria at the single-cell level and observed the oscillation of cell-length of bacteria in the presence of $Ag^+$ ions [35]. Time-lapse imaging is able to provide valuable information on individual cells that is inaccessible at the ensemble level. However, based on our experience, time-lapse microscopy is not as convenient as the microplate-reader based method in certain situations, especially when high throughputs with different growth conditions are needed. It has several limitations, including relatively lower throughput, only one condition at a time, defocusing over long time period, stage drift, limited size of field of view (FOV), and variance among different FOVs. It is worthwhile to note that there exist modern microplate readers that combines time-lapse imaging with growth curve measurements, which combines the advantages from both types of measurements.

As the current method makes use of time derivatives, it requires reasonably high temporal resolution in the measurements (i.e., the time intervals of data points should be much shorter than the growth dynamics of bacteria). Therefore, we expect that the current method is *not* suitable for CFU assays, which typically do not have good temporal resolutions. However, we believe that our method can be extended to other assays beyond OD and fluorescence. For example, it would be interesting to experimentally demonstrate the current method on ATP abundance-based assays using luminescence in the future. Another potential future study is to extend the time-derivative method to microbiological/biochemical assays for other purposes, such as examining the metabolic rate of microbes and enzymatic activities.

## Conclusions

To summarize, we developed a method for evaluating the growth of bacteria measured by multimode microplate readers. This method is based on the time derivatives of the absorption (OD) and/or fluorescence (FL) of the bacterial culture. Using quantitative models predicting the cell number $n$ and number of fluorescent proteins $p_f$ as functions of time, we characterized

the dependence of the first-order time derivative of OD and the second-order time derivative of FL on various parameters of the model, which are related to the commonly accepted lag-time and maximum specific growth rate of bacterial cultures. We showed that our method is consistent with traditional mathematical fittings of sigmoid growth curves (e.g., the Gompertz model). In addition, our method provides a better way for interpreting OD growth curves when multiple scattering is problematic and the growth curve does not follow the well-established sigmoid shape, which solves the issue of nonlinear relation between OD and cell number due to multiple scattering in growth curve measurements using microplate readers. More importantly, our method provides a framework for understanding the FL growth curves and extracting the growth behavior of bacteria from the FL measurements, which is especially useful when the absorption and scattering are affected by other components in the bacterial culture.

As a demonstration, we applied the current method for investigating the elongation of lag time for bacteria treated with $Ag^+$ ions and found that results from our method were in great agreement with the traditional mathematical fittings from the partial OD growth curves (at low values where the multiple scattering can be ignored). In addition, the current method was applied to the growth of bacteria in the presence of AgNPs at various concentrations, where the traditional growth measurement failed due to the high scattering, high absorption, and other processes from the AgNPs. Our method allowed us to successfully extract how the AgNPs affected the growth of bacteria.

## Supporting information

**S1 File.**
(PDF)

## Acknowledgments

We thank Dr. David McMillen at the University of Toronto for the generous gift of plasmid encoding the enhanced GFP and ampicillin resistance.

## Author Contributions

**Conceptualization:** Yong Wang.

**Data curation:** Venkata Rao Krishnamurthi, Isabelle I. Niyonshuti, Yong Wang.

**Formal analysis:** Venkata Rao Krishnamurthi, Yong Wang.

**Funding acquisition:** Jingyi Chen, Yong Wang.

**Investigation:** Venkata Rao Krishnamurthi.

**Methodology:** Yong Wang.

**Supervision:** Yong Wang.

**Validation:** Venkata Rao Krishnamurthi, Yong Wang.

**Visualization:** Venkata Rao Krishnamurthi, Yong Wang.

**Writing – original draft:** Venkata Rao Krishnamurthi, Yong Wang.

**Writing – review & editing:** Venkata Rao Krishnamurthi, Isabelle I. Niyonshuti, Jingyi Chen, Yong Wang.

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
