## [Decision Letter · Decision Letter 0]

4 Dec 2020

PONE-D-20-34200

A new analysis method for evaluating bacterial growth with microplate readers

PLOS ONE

Dear Dr. Wang,

Thank you for submitting your manuscript to PLOS ONE. After careful consideration, we feel that it has merit but does not fully meet PLOS ONE’s publication criteria as it currently stands. Therefore, we invite you to submit a revised version of the manuscript that addresses the points raised during the review process.

We look forward to receiving your revised manuscript.

Kind regards,

Ravi Pratap Barnwal, Ph.D.

Academic Editor

PLOS ONE

Journal Requirements:

Reviewers' comments:

Reviewer's Responses to Questions

**Comments to the Author**

1. Is the manuscript technically sound, and do the data support the conclusions?

Reviewer #1: Partly

Reviewer #2: Partly

Reviewer #3: Yes

2. Has the statistical analysis been performed appropriately and rigorously? 

Reviewer #1: Yes

Reviewer #2: Yes

Reviewer #3: N/A

3. Have the authors made all data underlying the findings in their manuscript fully available?

Reviewer #1: Yes

Reviewer #2: Yes

Reviewer #3: Yes

4. Is the manuscript presented in an intelligible fashion and written in standard English?

Reviewer #1: Yes

Reviewer #2: Yes

Reviewer #3: Yes

5. Review Comments to the Author

Reviewer #1: The manuscript by Krishnamurthi et al is an interesting work that relates to the general working in a microbiology laboratory. The method developed here allows one to estimate the lag time in the growth of a bacterium, which can be used for comparison between conditions such as, for example, normal and toxic. While the work and the analyses are generally good, there are several issues with the paper that I describe below.

Major comments:

1. Lines 58 - 60. “First, multiple scattering of microbes is more severe and problematic in microplate reader-based OD measurements than in cuvette-based ones (2)” - The reference given does not provide any information to support this statement. Multiple scattering is a function of cell density in the sample. The error introduced by multiple scattering, however, depends on factors such as aperture radius and the distance between detector and sample, etc. of the instrument used. The cited paper, however, does not suggest anything about how these factors vary between plate readers and conventional cuvette-based experiments. The authors only claim that since plate readers are predominantly used for measurements at higher culture densities, there are errors in estimating cell numbers, which, I believe, is true even for cuvette-based experiments.

2. Line 123. It is not clear why Pn (number of non-fluorescent proteins) is used in the numerator of the second part of the RHS of the equation “P/ = kP − P/(P+P+).” It should have been P instead of P since this part of the equation accounts for the degradation of fluorescent GFP proteins in the system.

3. Lines 223-224. “We note that there is a small horizontal shift between the two peaks (Fig. 2F)”. – Here, this shift between the peaks of OD and fluorescence derivate curves is not further explored, particularly if the shift is a function of any of the growth or fluorescent protein expression parameters. If it is so, comparing this peak location across samples where these parameters vary could produce errors.

4. Although the authors claim that the first derivative of growth curve can be used in calculating growth parameters from samples that do not fit into traditional sigmoid curves due to multiple scattering, none of the modelling results justify why this can be done. At high multiple scattering regimes, OD is expected to have a parabolic dependency on bacterial number, N (see your reference 2). It is not clear why taking first derivative should eliminate the non-linear cell concentration dependent contribution of multiple scattering to OD. None of the stimulations performed has included this contribution from multiple scattering. Calculating growth parameters from derivative curves of non-sigmoid growth profiles when the relationship between OD curve and its derivate itself was derived using a simple sigmoid growth model seems erroneous.

5. Since it is shown in the simulations that the peak location and peak height of the second derivative of fluorescence curves is dependent on parameters such as GFP expression rate, maturation rate, degradation rate, etc., it should be noted in the limitations part that the use of fluorescent readings can be used to estimate cell density across different samples only if these parameters for fluorescence protein concentration remain constant throughout the samples compared. One of the most important limitations of the proposed strategy could be that the value obtained from the second derivative graph of fluorescence, namely peak location is dependent on multiple independent growth factors such as activation rate and maximum growth rate. Therefore the peak location obtained from graphs of different samples can only be used to quantify the change in one of the growth parameters if one makes sure that the other parameter remains constant. This is not the case most of the time.

Other minor comments:

1. It is not clear how GFP expresses - from a constitutive promoter or induced? I was unable to find details regarding the plasmid even from the cited reference.

2. Is it possible that the GFP expression can directly be affected by the added Ag? That will also lead to errors in lag time calculation. Further, if added Ag affects plasmid replication, this will be an issue. In such scenario, it would be preferred to have an integrated plasmid expressing GFP.

3. One of the outcomes of the growth curve is the mean generation time. This is not possible with the method developed here.

Reviewer #2: The manuscript by Krishnamurthi et al showed a new time derivative method for evaluating the bacterial growth. The precise evaluation on the growth rate is highly required for better understanding of growing cells and populations. Authors have beautifully targeted and addressed a very the basic challenge pertaining to the evaluation of the bacterial growth. Some of the questions and queries that came across are as follows:

1. Authors should re-check the equation 4 on page 5 of the manuscript. In my opinion second term on right hand site should be degradation of number of fluorescent proteins. Or if there is any specific reason for inclusion of the said term, it should be specified.

2. Generally Optical density-based methods requires high bacteria concentrations (about 107 cells/ml). Since current method is based on the time derivatives of optical density, the robustness of the method needs to be mentioned in terms of bacterial concentrations especially at lower concentrations.

3. Further, considering the fact that more robust, time-lapse bright field imaging system are available that enable rapid higher throughput, non-invasive, real-time monitoring of microbial growth and morphological features, how current method resolves the challenge of differentiating the live and dead cells based on OD or FL.

4. How about the versatility of this model? Is this method applicable to other methods such as colony forming unit (CFU) and ATP abundance-based methods used to monitor bacterial growth?

5. Recently, Optical Density and fluorescence based more robust methods of measuring bacterial growth by simple calibrations using silica microsphere have been proposed (Beal, J., Farny, N.G., Haddock-Angelli, T. et al. Robust estimation of bacterial cell count from optical density. Commun Biol 3, 512 (2020). https://doi.org/10.1038/s42003-020-01127-5). What are the advantages of your method over these minute calibrations?

Minor Comments:

1. Conclusion section needs to be shortened

Reviewer #3: The authors in this study aims to develop a method for evaluating the growth of bacteria measured with multimode microplate readers based on the time derivatives of the OD and/or FL of the bacteria. Moreover, they conclude that their methods provides a better way for interpreting OD growth curves when the growth curve does not follow the well-established sigmoid shape due to multiple scattering. The work is good and the paper is will written. However, there are some minor points that need to be warranted before the publication of this manuscript.

• The authors have tested their hypothesis on bacteria expressing Green Fluorescent Protein. This implies that the method is applicable to only those bacterial population expressing GFP. If we are working with some pathogenic bacteria which don’t express GFP, then will this method work?

• Since this method works with bacteria having GFP, the authors assume that as bacterial cell divides, the GFP molecule will be passed on to the daughter cells. However, this is not true and many times plasmid is lost from daughter cells if they are not grown in selective media. Have the authors taken this into the account?

• During the growth phase of the bacteria, death of bacterial cells also occur. Does the authors have included death parameter as well in the equations and experimentation process?

• There are some reports linking the quality of GFP expression and bacterial growth. Have the authors taken account of this fact in their study?

6. PLOS authors have the option to publish the peer review history of their article (what does this mean?). If published, this will include your full peer review and any attached files.

Reviewer #1: No

Reviewer #2: No

Reviewer #3: No

---

## [Author Response · Author response to Decision Letter 0]

21 Dec 2020

Please see the uploaded "Response to Reviewers" document. Thank you.

---

## [Editor Report · Decision Letter 1]

26 Dec 2020

A new analysis method for evaluating bacterial growth with microplate readers

PONE-D-20-34200R1

Dear Dr. Wang,

We’re pleased to inform you that your manuscript has been judged scientifically suitable for publication and will be formally accepted for publication once it meets all outstanding technical requirements.

Kind regards,

Ravi Pratap Barnwal, Ph.D.

Academic Editor

PLOS ONE
---

## [Editor Report · Acceptance letter]

2 Jan 2021

PONE-D-20-34200R1 

A new analysis method for evaluating bacterial growth with microplate readers 

Dear Dr. Wang:

I'm pleased to inform you that your manuscript has been deemed suitable for publication in PLOS ONE. Congratulations! Your manuscript is now with our production department. 

Kind regards, 

on behalf of

Dr. Ravi Pratap Barnwal 

Academic Editor

PLOS ONE